# Expression of Two α-Type Expansins from *Ammopiptanthus nanus* in *Arabidopsis thaliana* Enhance Tolerance to Cold and Drought Stresses

**DOI:** 10.3390/ijms20215255

**Published:** 2019-10-23

**Authors:** Yanping Liu, Li Zhang, Wenfang Hao, Ling Zhang, Yi Liu, Longqing Chen

**Affiliations:** 1College of Horticulture & Forestry Science, Huazhong Agricultural University, Wuhan 430070, China; 2College of Life Science, Tarim University/Xinjiang Production and Construction Corps Key Laboratory of Protection and Utilization of Biological Resources in Tarim Basin, Alar 843300, China; haowenfang@nwsuaf.edu.cn (W.H.);; 3College of Life Science and Technology, Shenyang Agricultural University, Shenyang 110866, China; zhangli@syau.edu.cn; 4School of Chemical Engineering, Institute of Pharmaceutical Engineering Technology and Application, Sichuan University of Science & Engineering, Zigong 643000, China; liuyi0961@sina.com; 5Southwest Research Center of Engineering Technology for Landscape Architecture (State Forestry Administration), Southwest Forestry University, Kunming 650224, China

**Keywords:** expansin, *Ammopiptanthus nanus*, *AnEXPA1*, *AnEXPA2*, *Arabidopsis*, abiotic stress, hormone, reactive oxygen species

## Abstract

Expansins, cell-wall loosening proteins, play an important role in plant growth and development and abiotic stress tolerance. *Ammopiptanthus nanus* (*A. nanus*) is an important plant to study to understand stress resistance in forestry. In our previous study, two α-type expansins from *A. nanus* were cloned and named *AnEXPA1* and *AnEXPA2*. In this study, we found that they responded to different abiotic stress and hormone signals. It suggests that they may play different roles in response to abiotic stress. Their promoters show some of the same element responses to abiotic stress and hormones, but some special elements were identified between the expansins that could be essential for their expression. In order to further testify the reliability of the above results, we conducted an analysis of β-glucuronidase (GUS) dyeing. The analysis showed that *AnEXPA1* was only induced by cold stress, whereas *AnEXPA2* responded to hormone induction. *AnEXPA1* and *AnEXPA2* transgenic *Arabidopsis* plants showed better tolerance to cold and drought stresses. Moreover, the ability to scavenge reactive oxygen species (ROS) was significantly improved in the transgenic plants, and expansin activity was enhanced. These results suggested that *AnEXPA1* and *AnEXPA2* play an important role in the response to abiotic stress. Our research contributes to a better understanding of the regulatory network of expansins and may benefit agricultural production.

## 1. Introduction

Abiotic stresses, which are major limiting factors in agriculture, include conditions such as drought, salinity, high or low temperatures, light, deficient or excess nutrients, heavy metals, and pollutants. All of these factors, working individually or together, can endanger plants by negatively affecting their growth, development, and productivity [1]. The most common biological effect of the stresses is the restriction of plant growth through cell division and extension under stress conditions. However, the key problem in plant growth is the extension of cell regulation under various stresses [2]. Plant cell walls determine the shape of cells and help them resist biotic and abiotic stresses [3,4,5]. In addition, proteins in plant cell walls play a major role in the regulation of cell wall extension [3].

Expansins, a class of plant cell wall proteins, are mainly involved in inducing the extension of cell walls and regulating pH-dependent cell expansion [6]. The plant expansin protein family is classified into four subfamilies α-expansin, β-expansin, expansin-like protein A, and expansin-like protein B [7]. These proteins, which are involved in various physiological and biochemical pathways, are widely distributed in plants. For example, 37 expansin genes were found in the *Arabidopsis thaliana* (*A. thaliana*) genome. Among these, are 25 α-expansins, six β-expansins, three expansin-like proteins A, and three expansin-like proteins B; however, their functions are different [8,9,10,11]. In addition, 128 expansin genes have been found in the wheat genome [12], and many expansin proteins are involved in cold tolerance. Early reports revealed that the transcription activities of expansin proteins differ by tissue and organ type, and their functions are diverse [13,14,15]. Such studies were focused on fruit softening and interaction with plant cell walls [16,17]. Recently, researchers have noticed the relationship between expansins and abiotic stress [18,19]. Ectopic expressions of expansin genes were used to demonstrate their correlation, but the regulation model of expansin is still ambiguous. To understand this regulation model, the key is to determine the relationship among cell wall-loosening, expansins and abiotic stress [20].

Stress tolerance and the growth of plants are involved in hormone signaling transduction [21,22]. Hormones are induced by specific environmental signals and regulate plant physiological responses. Different hormones cause different effects during the growth, development, and defense responses of plants, both synergistically and antagonistically. Previous studies revealed that the expression of expansin could be induced by hormones. For example, researchers found that methyl jasmonate (MeJA) and 1-Naphthaleneacetic acid (NAA) could positively regulate *EgEXPA2* and *EgEXPA3* in the flower opening of *Eustoma* [23].

*Ammopiptanthus nanus* (*A. nanus*) is one of two species of *Ammopiptanthus* (Leguminosae). In China, it is only found in the northwestern desert region and is the only evergreen broad-leafed shrub in the desert. Moreover, the distribution of *A. nanus* is narrow; it spans across an altitude of 2100 to 2400 m. Thus, it is a rare species that classifies as first class in the protected category. It has important scientific value in determining the occurrence and development of the flora of desert zones in the southwest of Xinjiang. The habitat of *A. nanus* is poor, with a dry climate, low rainfall, and barren soil. In this area, the annual average temperature is 6.8 °C, and the temperature varies from −30 °C to 47.6 °C. Annual precipitation is 150–200 mm, and evaporation exceeds 2000 mm [24]. In our previous study, two α-type expansins from *A. nanus*, *AnEXPA1* and *AnEXPA2*, were found in a cold-induced RNA-seq library of *A. nanus* seedlings. They responded to different abiotic stresses and hormones. Their promoters showed several elements that responded to abiotic stress and hormone signals. Nevertheless, they responded differently to abiotic stresses and hormones. To understand their functions, *AnEXPA1* and *AnEXPA2* transgenic *Arabidopsis* plants were used to analyze tolerance to cold and drought stresses in this study.

## 2. Results

### 2.1. Cloning and Characterization of AnEXPA1 and AnEXPA2

Two *AnEXPA* gene coding sequence (CDS) fragments named *AnEXPA1* and *AnEXPA2* were obtained from a cold-induced RNA-Seq library of *A. nanus* seedlings (KC184696 and KC184697, respectively). Both genes had a 747-bp CDS sequence and encoded a polypeptide of 248 amino acids, with predicted molecular weights of approximately 26.53 and 26.69 kD, respectively. AnEXPA1 shared a high similarity with AnEXPA2 (86.92%). However, AnEXPA1 had a 62.86% similarity with AtEXPA15, 39.68% with AtEXPA24 from *Arabidopsis*, 69.21% with GmEXPA4 from *Glycine max*, 84.68% with MtrEXPA10 from *Medicago truncatula*, and 68.25% with GhEXPA10 from *Gossypium hirsutum*. In addition, AnEXPA2 had a 63.81%, 40.32%, 72.38%, 86.96%, and 68.25% similarity with these respective proteins. Among these homologous proteins, which had between 248 and 312 amino acids, there were five gaps, including four in the region of the N- or C-terminus. Thus, high diversity was found in the first and last 25 positions of the N-terminus and C-terminus respectively, except for AtEXPA24. In addition, both AnEXPA1 and AnEXPA2 had two domains; the first one (DPBB_1) had between 60 and 145 amino acids (Figure 1 domain I), and the second (Pollen_allerg_1, CBM63) domain ranged from amino acids 158 to 229 (Figure 1 domain II) [25]. The signal peptide of AnEXPA1 was from one to 22 amino acids long, with a cleavage site between 22 and 23 amino acids, but that of AnEXPA2 was from one to 18 amino acids long, with a cleavage site between 18 and 19 amino acids (Figure 1). The functional histidine, phenylalanine, and aspartate residue (HFD) motif similar to that of the family 45 of glycosyl hydrolases [26] was located between 123 and 125 amino acids. There were 33 differences in the 248 amino acids, of which eight amino acids had obvious differences in hydrophobicity between AnEXPA1 and AnEXPA2. Moreover, among these eight different amino acids, four were located in the two structural domains. Two of the four amino acids were located in the DPBB domain: A of AnEXPA1 and N of AnEXPA2 at the 84th amino acid, and A of AnEXPA1 and S of AnEXPA2 at the 144th amino acid. In the CBM63 domain, the different amino acids of AnEXPA1 (A) and AnEXPA2 (V) were located in the 183rd position, whereas S (AnEXPA1) and P (AnEXPA2) were at the 194th position. Therefore, the different amino acids of these two genes could lead to differences in their protein structure and function [25,27,28].

### 2.2. AnEXPA1 and AnEXPA2 Showed Similar 3D Structure

To explore the structural difference of the AnEXPA1 and AnEXPA2 proteins, the software WebLab ViewerLite 4.0 (Molecular Simulations Inc., CA, USA) was used to construct a 3D model. In order to analyze protein spatial conformational differences, two different templates of beta-expansin 1a (2hcz.1.A) and pollen allergen (Phlp 1, 1n10.1.A) were used to build 3D protein models of these two proteins. As Appendix A shows, the dark blue represents the N end of the protein, and the deep red represents the C end. There was 32.71% similarity between AnEXPA1 and beta-expansin 1a and 30.52% similarity between AnEXPA1 and Phlp 1 protein. For AnEXPA2, the similarity degree between the two templates was 32.85% and 31.73%, respectively (Appendix A). AnEXPA1 had two conserved domains, with only one α-helix in domain I, but the number of β-sheets was different for the two templates. There were 10 (Appendix A) and 11 (Appendix A) β-sheets in domain I, whereas there were nine (Figure 3A) and 11 (Appendix A) β-sheets in domain II of AnEXPA1. AnEXPA2 also had two conserved domains with one α-helix in domain I. For the β-sheet, there were 10 (Appendix A) and nine (Appendix A) β-sheets in domain I, whereas both proteins had 10 β-sheets in domain II (Appendix A). Domain II of these two proteins had a cylindrical shape; this structure might be involved in protein functioning. In domain I, only one α-helix was located on its outboard, and β-sheets were below this α-helix. Thus, this different structure could determine protein function.

### 2.3. Phylogenetic Analysis

To determine the evolutionary relationships between these two genes and other plant expansin proteins, AnEXPA1 and AnEXPA2 amino acid sequences were analyzed with 59 other expansin proteins. *Ammopiptanthus nanus* is a Leguminosae plant, therefore some expansin proteins from the Leguminosae species were first selected for comparison with the AnEXPA1 and AnEXPA2 sequences, such as *Glycine max*, *Arachis ipaensis*, *Arachis duranensis*, *Medicago truncatula*, and *Vigna angularis*. Moreover, expansin proteins from the model plant *Arabidopsis thaliana* were also selected for comparison with AnEXPA1 and AnEXPA2 sequences. In addition, some proteins with a high similarity with AnEXPA1 and AnEXPA2 (more than 60%) were also used for classification, such as Malvaceae (*Gossypium hirsutum*), Salicaceae (*Populus euphratica*), Stellariaceae (*Theobroma cacao*), Sapindaceae (*Dimocarpus longan*), Pinus (*Pinus taeda*), Taxodiaceae (*Cunninghamia lanceolata*), and Nelumbonaceae (*Nelumbo nucifera*). Among these proteins, AtEXPA24 was the longest; it had 312 amino acids. The shortest was AtEXPA14, with 203 amino acids. These expansin proteins could be classified into three major clades (α-expansin, β-expansin, and expansin-like). AnEXPA1 and AnEXPA2 were classified as clade α-expansins with 46 other expansin proteins from 15 species (Figure 2). This suggested that they might have a similar function.

### 2.4. Cis-Element Analysis of AnEXPA1 and AnEXPA2 Promoters

The promoter fragments of *AnEXPA1* and *AnEXPA2*, with 1163 bp and 1033 bp, respectively, were cloned (KX247396 and KX354938) (Appendix A). The online software PLACE was used to analyze the cis-acting elements of the promoters. A prediction of the cis-acting elements showed that *AnEXPA1* and *AnEXPA2* (Appendix A) had six and eleven cis-acting elements that related to stress and hormone responses, respectively. Six stress responses (dehydration and cold) and hormones (gibberellin, auxin, and salicylic acid) were found in the *AnEXPA1* promoter, including MYB1AT (WAACCA), MYCCONSENSUSAT (CANNTG), ASF-1 binding site (TGACG), PYRIMIDINEBOXOSRAMY1A (CCTTTT), MYBCORE (CNGTTR), and GAREAT (TAACAAR). Among the six elements, MYB1AT, MYCCONSENSUSAT, PYRIMIDINEBOXOSRAMY1A, and MYBCORE also existed in the *AnEXPA2* promoter (Appendix A). These results suggested that the four common cis-acting element binding transcription factors may regulate expression of *AnEXPA1* and *AnEXPA2*. In addition, the *AnEXPA2* promoter showed several elements response to ABA (CATGCA; CACATG; CTAACCA), GA (TGAC), CBF (RYCGAC), dehydration (CATGTG), and the auxin (CATATG) signal. These elements specifically existed in the *AnEXPA2* promoter; elements that responded to auxin and/or salicylic acid, abiotic and biotic stress (TGACG), and the GA-responsive element (TAACAAR) were only found in the *AnEXPA1* promoter (Appendix A). These different elements may affect *AnEXPA1* and *AnEXPA2* expression when exposed to abiotic stress and hormone signals.

### 2.5. AnEXPA1 and AnEXPA2 Response to Different Abiotic Stresses and Hormone Induction

To explore the effects of abiotic stress on *AnEXPA1* and *AnEXPA2* expression, the *A. nanus* seedlings were treated with 4 °C, 20% PEG6000, and 0.25 M NaCl, and the expressions of *AnEXPA1* and *AnEXPA2* were analyzed in the roots, stems and leaves. In *A. nanus* roots, *AnEXPA1* expression was increased 7.48-fold at 4 °C for 18 h, and *AnEXPA2* expression was up-regulated by 10.74-fold at 4 °C for 24 h (Figure 3A). However, in the *A. nanus* stems and leaves, *AnEXPA1* expression was up-regulated by cold stress, in contrast to *AnEXPA2* (Figure 3B,C). These results suggested that both *AnEXPA1* and *AnEXPA2* are induced by cold stress, but their expressions are different in different tissues. In addition, *AnEXPA2* expression was decreased in all tissues, but *AnEXPA1* expression was increased in the roots and stems under drought and salt stresses (Figure 3D,E,G,H) and did not change in the leaves (Figure 3F,I). This suggested that drought and salt stresses repress *AnEXPA2* expression but improve *AnEXPA1* expression.

To explore the effects of hormones on *AnEXPA1* and *AnEXPA2* expression, the *A. nanus* seedlings were treated with 50 μM gibberellin (GA3), 50 μM ethephon (ET), 1 μM indole-acetic acid (IAA), 2 μM NAA, 2 μM abscisic acid (ABA), and 10 μM MeJA, and the expressions of *AnEXPA1* and *AnEXPA2* were analyzed in the roots, stems, and leaves. During several hormone treatments, *AnEXPA1* expression was only increased by IAA and NAA (Figure 4C,D), but decreased by GA3, ET, and MeJA (Figure 4A,B,F). In addition, the ABA treatment did not affect *AnEXPA1* expression (Figure 4E). These results suggested that *AnEXPA1* expression was positively correlated with IAA and NAA, negatively correlated with GA3, ET, and MeJA, and not correlated with ABA. However, *AnEXPA2* expression was increased by GA3, ET, IAA, ABA, and MeJA (Figure 4A–C,E,F), and was not affect by NAA (Figure 4D). This suggested that GA3, ET, IAA, ABA, and MeJA were positively correlated with *AnEXPA2* expression, but NAA did not work. These results suggested that the expression of the two expansin genes showed tissue specificity and responded to different abiotic stresses and hormones.

### 2.6. Promoters of AnEXPA1 and AnEXPA2 Showed Different Effects during Abiotic Stress and Hormone Induction

To further verify the reliability of the qRT-PCR results during different abiotic stress and hormone treatments, tobacco leaves with *proAnEXPA1::GUS* and *proAnEXPA2::GUS* were used to analyze β-glucuronidase (GUS) activity. Under normal conditions, both the promoters of *AnEXPA1* and *AnEXPA2* activated the transcription of the GUS reporter gene, but the production of GUS was low (Figure 5A). However, the promoter of *AnEXPA1* did not improve GUS production under 40% PEG6000 and 0.25 M NaCl treatment (Figure 5B,C), but they increased GUS production under 4 °C stress (Figure 5D). These results suggested that *AnEXPA1* significantly responds to cold stress but not to drought and salt stress. In tobacco leaves with *proAnEXPA2::GUS*, drought, salt, and cold stresses did not induce, but instead reduced GUS production (Figure 5B–D). Moreover, the GUS activity assay showed the same results (Figure 5H,I). This indicated that *AnEXPA2* may not be involved in the response to abiotic stress.

After GA3, ABA, and IAA treatment, the GUS protein was not significantly increased in the tobacco leaves with *proAnEXPA1::GUS*, but it was significantly increased in leaves with *proAnEXPA2::GUS* (Figure 5E–G). In addition, this phenomenon was verified by a GUS activity assay after three hormone treatments (Figure 5H,I). These results suggested that *AnEXPA2* may be involved in the hormone signaling pathway, but that *AnEXPA1* is not responsive to hormone signals.

### 2.7. Overexpression of AnEXPA1 and AnEXPA2 in Arabidopsis Enhanced Tolerance to Cold Stress

In this study, we obtained six *AnEXPA1* and 10 *AnEXPA2* transgenic *Arabidopsis* lines. Among these lines, *AnEXPA1* transgenic lines #17 and #28, and *AnEXPA2* transgenic lines #5 and #22 were single-copy-number homozygotes. Thus, these four transgenic lines were used in this study. When *AnEXPA1* and *AnEXPA2* transgenic plants were treated at 6 °C for 8 d, not all transgenic plants showed obvious damage (Figure 6A,B). However, the growth rate of the non-transgenic plants (WT) was slower than that of *AnEXPA1* and *AnEXPA2* (Figure 6B), indicating that the overexpression of *AnEXPA1* and *AnEXPA2* enhances the cold tolerance of transgenic plants and improves plant growth during cold stress. After 6 °C for 3 days, the electrical conductivity, malondialdehyde (MDA), and H_2_O_2_ content in all the transgenic lines were significantly lower than that in the WT lines (Figure 6C,D,H). This suggested that overexpression of *AnEXPA1* and *AnEXPA2* decreases reactive oxygen species (ROS) content. In addition, superoxide dismutase (SOD) activity was increased in all *AnEXPA1* and *AnEXPA2* transgenic lines under normal and low temperature (Figure 6E). This suggested that the overexpression of *AnEXPA1* and *AnEXPA2* enhances the ROS scavenging ability by increasing SOD activity. Although our results verified that the cold tolerance of all transgenic lines was enhanced, the activities of peroxidase (POD) and catalase (CAT) were not improved under a normal and low temperature (Figure 6F,G). This could be due to constitutive high SOD protein levels, which may have led to low levels of ROS production.

### 2.8. Overexpression of AnEXPA1 and AnEXPA2 Enhanced Transgenic Plant Tolerance to Drought Stress

To explore whether overexpression of *AnEXPA1* and *AnEXPA2* enhances the drought tolerance of all transgenic lines, all transgenic lines were treated with water shortage. After controlling water for 7 days, most of the WT plants were dead, and almost all the leaves had withered (Figure 7A), suggesting that non-transgenic *Arabidopsis* plants do not bear long-term water deficiency. However, *AnEXPA1* transgenic lines #17 and #28, as well as *AnEXPA2* transgenic lines #5 and #22 showed strong drought tolerance after controlling water for 7 d (Figure 7A). These results suggested that both *AnEXPA1* and *AnEXPA2* are involved in the drought tolerance of plants. To understand the effects of *AnEXPA1* and *AnEXPA2* on scavenging ROS, antioxidant enzyme activity and antioxidant contents were analyzed during drought stress. At normal water conditions and water control for 1 d, all *AnEXPA1* and *AnEXPA2* transgenic lines showed similar electrical conductivity, contents of MDA and H_2_O_2_, and activities of SOD, POD, and CAT (Figure 7B–G). These results suggested that the antioxidant ability of *AnEXPA1* and *AnEXPA2* transgenic lines were not affected by short-term water deficiency treatment. After controlling water for 3 d, the electrical conductivity and H_2_O_2_ contents in the *AnEXPA1* transgenic lines were lower than those in the WT plants; the MDA content activities of SOD, POD, and CAT were not changed. In addition, overexpression of *AnEXPA2* only decreased the H_2_O_2_ content; it did not affect electrical conductivity, MDA content, or the activities of SOD, POD, and CAT. These results suggested that *AnEXPA1* and *AnEXPA2* work on scavenging ROS under water shortage for 3 days. After controlling water for 7 d, the electrical conductivity and the content of MDA and H_2_O_2_ in all *AnEXPA1* and *AnEXPA2* transgenic lines were significantly lower than those in the WT plants. Moreover, the activities of SOD, POD, and CAT were significantly increased in all transgenic lines (Figure 7B–G). This suggested that the effects of *AnEXPA1* and *AnEXPA2* in scavenging ROS were gradually enhanced when extending the treatment time. In addition, the expansin activities of all transgenic plants were significantly enhanced under cold and drought stresses (Figure 7H). These results suggested that the *AnEXPA1* and *AnEXPA2* proteins also play a role in improving the cold and drought tolerance of transgenic plants by regulating cell-wall-loosening.

## 3. Discussion

Expansins, a class of important cell wall proteins, are widely distributed among angiosperms, nonflowering plants, algae, bacteria, and fungi in nature [29,30,31]. In plants such as wheat, maize, and soybean, the number of expansin genes can reach dozens [12]. Expansins are involved in many physiological and biochemical processes, particularly in the construction of cell walls. They also play an important role in plant growth and development as well as tolerance to abiotic stress, either alone or through interactions with other expansins.

### 3.1. AnEXPA1 and AnEXPA2 Responded to Different Abiotic Stresses and Hormone Signals

Gene expression is a complex process that involves in the interaction of cis-acting elements with trans-factors. Some promoter elements may be important molecular switches that regulate gene transcription activity during various biological processes, such as development and response to abiotic stress and hormones [32,33,34]. Moreover, different expansins respond to different abiotic stresses and hormones. For example, tobacco *NtEXPA6* was not responsive to abiotic stress or hormone signals, but *NtEXPA1*, *NtEXPA4*, and *NtEXPA5* were induced by several stresses and the ABA signal [35]. In addition, wheat *TaEXPA8A*, *TaEXPA8B*, and *TaEXPA8D* responded to cold and drought stresses. The *TaEXPA8B* and *TaEXPA8D* transgenic plants showed stronger tolerance to cold stress, increased the activities of SOD, POD, and CAT, and improved the contents of soluble protein, MDA, and proline [36]. The analysis of cis-acting elements in the promoter regions of *AnEXPA1* and *AnEXPA2* genes revealed elements related to abiotic stress and hormone responses, such as CBF/DREB1, GA, and ABA (Appendix A). Although promoters of *AnEXPA1* and *AnEXPA2* showed some of the same elements that responded to abiotic stress and hormones, such as dehydration-responsive, CBF/DREB1, GA, and water stress, *AnEXPA1* was induced by cold, drought, and salt stresses in all *A. nanus* tissues, and only responded to NAA treatment. Moreover, *AnEXPA2* was only increased in cold-induced roots, and was negatively regulated in other tissues by drought and salt stresses. These results suggested that these same elements in their promoters may not interact with the correlated transcription factor. In contrast, the special elements in their promoters may play important roles in response to abiotic stress and hormone signals.

Two elements that were responsive to auxin and/or salicylic acid, abiotic and biotic stress (TGACG), and the GA-responsive element (TAACAAR), were found in the *AnEXPA1* promoter, which did not exist in the *AnEXPA2* promoter (Appendix A). However, the promoter of *AnEXPA2* showed several special elements, such as three ABA (CATGCA; CACATG; CTAACCA), GA (TGAC), CBF (RYCGAC), as well as dehydration (CATGTG) and auxin (CATATG) response elements, which did not exist in the *AnEXPA1* promoter. Reports indicated that the ABA-responsive CATGCA is involved in the cross-talk of plant hormones during rhizome development, storage protein, and the response to iron deficiency stress [37], whereas the GA element TGAC responds to wounding stress [38,39]. In addition, the TGACG binding factor not only regulates salicylic acid and pipecolic acid biosynthesis, but also affects hypocotyl elongation in soybean seedlings and drought response in foxtail millet seedlings [40,41,42]. These reports suggested that the cis-acting element interacts with different transcription factors during plant growth, development, and tolerance to abiotic stress. In this study, qRT-PCR, transient expression, and GUS activity assays indicated that *AnEXPA1* is mainly responsive to abiotic stress, and *AnEXPA2* is mainly involved in hormone signaling pathways. The results also suggested that these special cis-acting elements of their promoters play important roles in the response to abiotic stresses or hormone signals.

### 3.2. Overexpression of AnEXPA1 and AnEXPA2 in Arabidopsis Enhanced Tolerance to Cold and Drought Stresses

Expansins are key regulators of cell-wall loosening that are involved in plant tolerance to abiotic stress. Previous reports revealed that the overexpression of wheat *TaEXPB23* improved not only root growth and water stress, but also photosynthetic rate and antioxidant enzyme activity and decreased ROS accumulation in transgenic tobacco plants [43]. Moreover, overexpression of *AstEXPA1* from *Agrostis stolonifera* in tobacco enhanced tolerance to drought, heat, cold, and salt stresses in transgenic plants. In addition, *AstEXPA1* increased the content of soluble sugar and proline, and decreased relative electrolyte leakage and MDA content under heat, drought, and salt stresses [44]. In this study, overexpression of *AnEXPA1* and *AnEXPA2* in *Arabidopsis* enhanced the cold and drought tolerance of transgenic plants. Moreover, electrical conductivity and MDA and H_2_O_2_ content were significantly reduced, whereas SOD activity was increased in *AnEXPA1* and *AnEXPA2* transgenic plants under cold stress. However, POD and CAT activities were decreased in all transgenic plants under cold stress. The reason might be that the overexpression of *AnEXPA1* and *AnEXPA2* significantly decreased the level of the superoxide anion by improving SOD activity and its self-defense function. In addition, overexpression of *AnEXPA1* and *AnEXPA2* enhanced the cell-wall loosening ability of transgenic plants under cold stress. Responses to these defense systems further improve tolerance to cold stress in *AnEXPA1* and *AnEXPA2* transgenic plants.

Under osmotic stresses, such as drought and salt stresses, the level of the superoxide anion directly affects plant growth and development. Many reports indicated that a high level of expansins enhances ROS-scavenging ability. For example, the overexpression of wheat *TaEXPA2* improved oxidative stress tolerance in transgenic *Arabidopsis* plants and increased the expression of *AtPOD31*, *AtPOD33*, *AtPOD34*, *AtPOD71*, and POD activity under H_2_O_2_ stress [45]. A gene silencing assay verified that a low level of barley *HvEXPB7* suppressed root hairs under normal and drought stress and decreased K uptake [46]. These reports indicated that expansins are involved in regulating many physiological and biochemical processes under normal and abiotic stress, such as the expression of cell wall POD genes to scavenge ROS, and via ion uptake. In this study, the overexpression of *AnEXPA1* and *AnEXPA2* not only significantly enhanced tolerance to drought stress in transgenic plants, but also decreased electrical conductivity and MDA and H_2_O_2_ content under drought stress. Moreover, activities of SOD, POD, and CAT in all transgenic plants were increased under drought stress. These results suggested that high levels of *AnEXPA1* and *AnEXPA2* improve the ability to scavenge ROS during drought stress by regulating the activity of antioxidant enzymes. In addition, high levels of *AnEXPA1* and *AnEXPA2* enhanced cell-wall loosening ability under drought stress, suggesting that *AnEXPA1* and *AnEXPA2* work not only by regulating the antioxidant system, but also through the proteins affecting cell-wall loosening under drought stress.

In conclusion, *AnEXPA1* and *AnEXPA2* showed different expression models under abiotic stresses and hormone signals, suggesting that they play different roles in regulating *A. nanus* growth, development, and abiotic stress response. Although they responded to different abiotic stresses and hormone signals, the overexpression of *AnEXPA1* and *AnEXPA2* in *Arabidopsis* enhanced tolerance to cold and drought by improving ROS-scavenging ability and participating in cell-wall loosening. This research contributes to a better understanding of the stress tolerance of *A. nanus* and may benefit agricultural production.

## 4. Materials and Methods

### 4.1. Plant Materials

The *A. nanus* seeds used in this study were selected from Wuqia County, Xinjiang Uygur Autonomous Region, China. The seeds were sown in a Murashige and Skoog (MS) solid medium after surface sterilization. The seedlings were then placed in a tissue culture room and grown under 25 °C (16-h light/8-h dark) for one month before further treatment. One-month-old seedlings were treated for different time periods (0, 6, 12, 18, 24, and 48 h) with 4 °C, 0.25 M NaCl or 40% PEG6000. The different tissues were then harvested to analyze gene expression under abiotic stress. In addition, the seedlings were sprayed with 1 μM IAA, 2 μM abscisic acid (ABA), 2 μM NAA, 10 μM MeJA, and 50 μM GA3, and ET released from 50 μM ethephon. The seedlings were then cultured for different durations (0, 2, 8, 12, and 24 h) and were examined to determine the effects of different hormones on gene expression. After treatment, the seedlings were frozen in liquid nitrogen and conserved at −80 °C for total RNA extraction. The primers used in this study are shown in Appendix A.

### 4.2. RNA and DNA Extraction

The total RNA for all samples was extracted using a Trizol reagent (Invitrogen, Carlsbad, CA, USA). The isolated total RNA of all treated materials was used as a template, and cDNA was synthesized with M-MLV reverse transcriptase (Promega, Madison, WI, USA) for qRT-PCR. The RNA of the low temperature treatment was used to amplify the full-length cDNA. Then, the genomic DNA was isolated from the young leaves of *A. nanus* seedings with a Plant Genomic DNA Kit (Tiangen, Beijing, China). Lastly, the DNA was dissolved in appropriate volumes of Tris-HCl (TE) buffer for Genome Walking.

### 4.3. Genome Walking

The promoters and DNA sequences of *AnEXP1* and *AnEXP2* were isolated using the Universal Genome Walker^TM^ 2.0 kit (Clontech, Mountain view, CA, USA). The gene specific primers were designed according to the sequence of full-length cDNA using primer 5.0. The DNA sequences were spliced using DNASTAR software 7.1 (Madison, WI, USA).

### 4.4. Generation of Plant Expression Vectors

To understand the effects of abiotic stress and phytohormones on *AnEXP1* and *AnEXP2* expressions, the promoters of *AnEXP1* and *AnEXP2* were cloned and replaced the 35S promoter before the GUS reporter gene in the pCAMBIA3301 expression vector (Primers in Appendix A). The two recombinant plasmids were named *proAnEXP1::GUS* and *proAnEXP2::GUS* and were used in a transient expression experiment. To ensure a stable expression vector construction, the full-length coding sequences (CDS) of *AnEXPA1* and *AnEXPA2* were amplified by RT-PCR from *A. nanus* seedling leaves (Primers in Appendix A). The PCR product was purified and inserted into a pCAMBIA3301 expression vector using the In-Fusion HD Cloning Plus enzyme (Clontech, Mountain view, CA, USA), creating recombinant plasmids *35S::AnEXPA1-GUS* and *35S::AnEXPA2-GUS*.

### 4.5. GUS Dyeing and Protein Activity Assays

The *proAnEXP1::GUS* and *proAnEXP2::GUS* recombinant plasmids were electroporated into an *Agrobacterium* strain LBA4404. The bacterial liquid, including the recombinant plasmid (OD = 0.6), was injected into young tobacco leaves. Then, all tobacco plants were placed in the dark at 25 °C for 2 days, and were then cultured under 16-h light/8-h dark at 25 °C for 2 days. These tobacco plants were treated with 4 °C, 40% PEG6000 or 0.25 M NaCl for 24 h. In addition, a part of the tobacco plants was sprayed with 1 μM IAA, 2 μM ABA, and 50 μM GA3 and then cultured under 16-h light/8-h dark at 25 °C for 24 h. Finally, fresh leaves with the recombinant plasmid were dyed with a 2 mM X-Gluc solution (50 mM PBS, pH 7.0, 50 mM Na_2_HPO_4_, 10 mM Na_2_EDTA, 0.1% Triton 100, 0.5 mM K_4_[Fe(CN)_6_] and 2 mM X-beta-d-glucuronide cyclohexylammonium salt (X-Gluc)), and protein was extracted using an extraction buffer (50 mM sodium phosphate, pH 7.0). GUS activity was then determined fluorometrically using 4-methylumbelliferyl β-d-glucuronide as a substrate [47]. The production of 4-methylumbelliferone (MU) was measured using a fluorometer (CytoFluor, Applied Biosystems, FosterCity, CA, USA).

### 4.6. Arabidopsis Transgenic Plant Isolation

The *35S::AnEXPA1-GUS* and *35S::AnEXPA2-GUS* recombinant plasmids were electroporated into an *Agrobacterium* strain LBA4404. A positive *Agrobacterium* (OD = 1.0) was used to dip the unopened flower buds of *Arabidopsis*. T0 transgenic seedlings were selected using glyphosate to obtain positive transgenic plants. These positive transgenic lines were selected (three generations) until homozygous transgenic plants were obtained. The homozygous transgenic lines *AnEXPA1#17*, *AnEXPA1#28*, *AnEXPA2#5*, and *AnEXPA2#22* were used in this study.

### 4.7. Cold and Drought Tolerance of Transgenic Plant Assays

To investigate the functions of *AnEXPA1* and *AnEXPA2*, five-week-old seedlings of the transgenic lines *AnEXPA1#17*, *AnEXPA1#28*, *AnEXPA2#5*, and *AnEXPA2#22* were treated at constant temperature of 6 °C for 8 d (16-h light / 8-h dark) and were subjected to a water control treatment at 22 °C for 7 days (16-h light / 8-h dark). The wild-type *Arabidopsis* (WT) was used as a control. For MDA measurement, plant leaves (100 mg) were ground on ice in 5 mL of a 50-mM potassium phosphate buffer (pH 7.8). Homogenates were centrifuged at 4 °C for 12,000 *g* for 5 min. The supernatant (1 mL) was reacted with 1 mL 0.1% trichloroacetic acid (TCA) at 100 °C for 30 min, and then the values at 440, 532, and 600 nm were detected. Finally, the MDA content was calculated to reflect the lipid peroxidation. Fresh leaves (0.1 g) were cut into 0.1-cm fragments, placed in 25 mL ultrapure water, and evacuated for 10 min. The electrical conductivity was detected at 25 °C and 100 °C. In addition, 100 mg of leaves were ground in 5 mL of normal saline to extract crude enzymes. The supernatant (0.5 mL) was utilized to detect the H_2_O_2_ content, superoxide dismutase (SOD), peroxidase (POD), and catalase (CAT) activities [18,48]. Then, expansin activity was measured according to Zhou et al. [19].

### 4.8. Bioinformatics Analysis

Gene sequence analysis and the prediction of protein-conserved domains were performed on the NCBI server (http://www.ncbi.nlm.nih.gov/BLAST, http://www.ncbi.nlm.nih.gov/Structure/cdd/wrpsb.cgi). The SignalP 4.1 Server was used to predict protein signal peptides (http://www.cbs.dtu.dk/services/SignalP/). Then, the 3D model of the protein was constructed using WebLab ViewerLite 4.0. The Promoter analysis was conducted using PLACE (http://www.dna.affrc.go.jp/PLACE/signalsacan.html). DNASTAR software was used to translate the open reading frame (ORF) and to calculate the molecular weight of the deduced protein. The ClustalW [49] and GeneDoc [50] programs were used to analyze the multiple sequence alignments and phylogenetic trees. In addition, a phylogenetic tree was constructed using MEGA 4.0 software (Center for Evolutionary Medicine and Informatics, Tempe, AZ, USA) [51]. The structural analysis of the AnEXPA1 and AnEXPA1 proteins was accomplished using SWISS-MODEL [52,53] on the website http://www.expasy.org.

### 4.9. qRT-PCR

The qRT-PCR was conducted with the Real Master Mix (SYBR Green) (Tiangen, Beijing, China) in a total volume of 20 μL on an Applied Biosystems 7500 real-time PCR system (USA) according to the manufacturer’s instructions under the following conditions: 95 °C for 1 min, 40 cycles of 95 °C for 15 s, 60 °C for 30 s, and 68 °C for 1 min, 1 cycle of 95 °C for 15 s, 60 °C for 1 min, and 95 °C for 15 s. Primers for qRT-PCR were designed using primer 5.0 software according to the sequence of full-length cDNA. All primer sequences are shown in Appendix A. The primers of *Actin* were used to amplify the housekeeping gene as an internal control. Each reaction was repeated three times, and gene expression levels were calculated according to the method described by Pfaffl [54]. Significant differences in the experimental data were analyzed using Duncan’s multiple range tests.

### 4.10. Statistical Analysis

All experiments were repeated at least three times. Statistical analysis was conducted using the procedures of the data processing system (7.05, DPS; Zhejiang University, Zhejiang, China). Significant difference was tested at *p* < 0.05, 0.01 or 0.001 probability levels. The data were presented as the mean ± standard error (SE) of three independent experiments.

## Figures and Tables

**Figure 1 ijms-20-05255-f001:**
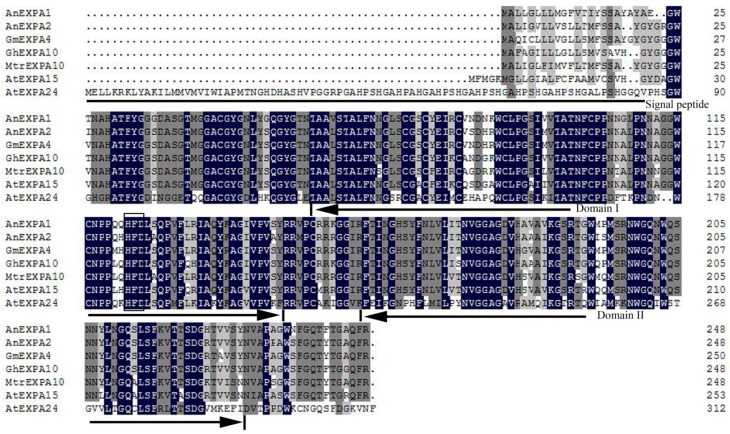
Alignment of AnEXPA1 and AnEXPA2 amino acid sequences with GmEXPA4 (NP_001240136), GhEXPA10 (XP_016674217), MtrEXPA10 (XP_003613849), AtEXPA15 (NP_178409), and AtEXPA24 (NP_198747). The signal peptide site is marked with a black line segment. The starting and ending sites of Domain I and II are marked with a single-headed arrow. The histidine, phenylalanine, and aspartate residue (HFD) sites are marked with a rectangular frame. Dark blue, similarity up 100%; dark gray, similarity up 75%; light gray, similarity up 50%.

**Figure 2 ijms-20-05255-f002:**
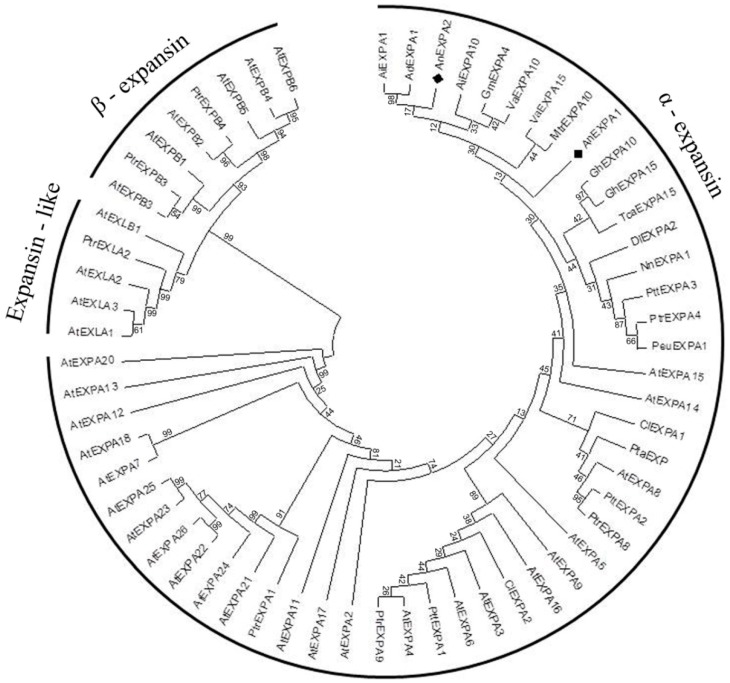
Phylogenetic analysis of AnEXPA1 and AnEXPA2 proteins. Evolutionary relationships were inferred using the neighbor-joining method. The bootstrap consensus tree inferred from 1000 replicates was used to represent the evolutionary history of the taxa analyzed. The Expansin proteins used in the phylogenetic tree analysis were from the plants *Ammopiptanthus nanus* AnEXPA1 (KC184696), AnEXPA2 (KC184697), *Gossypium hirsutum* GhEXPA10 (XP_016674217), GhEXPA15 (XP_016705834), *Arachis ipaensis* AiEXPA1 (XP_016190918), AiEXPA10 (XP_016202753), *Arachis duranensis* AdEXPA1 (XP_015956912), *Medicago truncatula* MtrEXPA10 (XP_003613849), *Populus euphratica* PeuEXPA1 (XP_011013147), *Theobroma cacao* TcaEXPA15 (XP_017971673), *Dimocarpus longan* DlEXPA2 (ACA05165), *Arabidopsis thaliana* AtEXPA2 (NP_196148), AtEXPA3 (NP_181300), AtEXPA4 (NP_181500), AtEXPA5 (NP_189545), AtEXPA6 (NP_180461), AtEXPA7 (NP_172717), AtEXPA8 (NP_181593), AtEXPA9 (NP_195846), AtEXPA11 (NP_173446), AtEXPA12 (NP_001327548), AtEXPA13 (NP_566197), AtEXPA14 (NP_001332570), AtEXPA15 (NP_178409), AtEXPA16 (NP_191109), AtEXPA17 (NP_192072), AtEXPA18 (NP_176486), AtEXPA20 (NP_195534), AtEXPA21 (NP_198742), AtEXPA22 (NP_198743), AtEXPA23 (NP_198744), AtEXPA24 (NP_198747), AtEXPA25 (NP_198746), AtEXPA26 (NP_198745), AtEXPB1 (NP_001324426), AtEXPB2 (NP_564860), AtEXPB3 (NP_567803), AtEXPB4 (NP_182036), AtEXPB5 (NP_001319806), AtEXPB6 (NP_001117554), AtEXLA1 (NP_190183), AtEXLA2 (NP_195553), AtEXLA3 (NP_001030815), AtEXLB1 (NP_193436), *Pinus taeda* PtaEXP (AF085330_1), *Populus tremula* × *Populus tremuloides* PttEXPA1 (AAR09168), PttEXPA2 (AAR09169), PttEXPA3 (AAR09170), *Populus trichocarpa* PtEXPB3 (XP_006371575), PtEXLA2 (XP_002313682), PtrEXPA1 (XP_002323897), PtrEXPA4 (XP_002315043), PtrEXPA8 (XP_002325925), PtrEXPA9 (XP_002315218), PtrEXPB4 (XP_002320708), *Cunninghamia lanceolata* ClEXPA1 (ABL09849), ClEXPA2 (ABM54492), *Glycine max* GmEXPA4 (NP_001240136), *Vigna angularis* VaEXPA10 (XP_017427959), VaEXPA15 (XP_017421014), and *Nelumbo nucifera* NnEXPA1 (XP_010258310). The two AnEXPA locations are marked with a black diamond.

**Figure 3 ijms-20-05255-f003:**
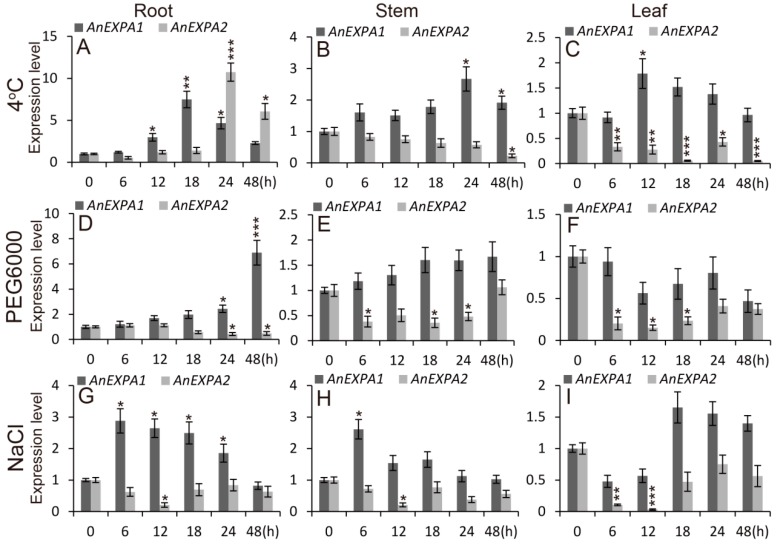
Expression levels of *AnEXPA1* and *AnEXPA2* during abiotic stress. (**A**–**I**) Relative expression levels of *AnEXPA1* and *AnEXPA2* in the roots, stems, and leaves of *A. nanus* seedlings during 4 °C, 20% PEG6000, and 0.25 M NaCl treatments. Relative amounts were calculated and normalized with respect to actin RNA, and the experiment was repeated at least three times. Data correspond to the mean ± standard error (SE) of three independent replicates. Different symbols above the columns indicate significant differences (* *p* < 0.05, ** *p* < 0.01, *** *p* < 0.001).

**Figure 4 ijms-20-05255-f004:**
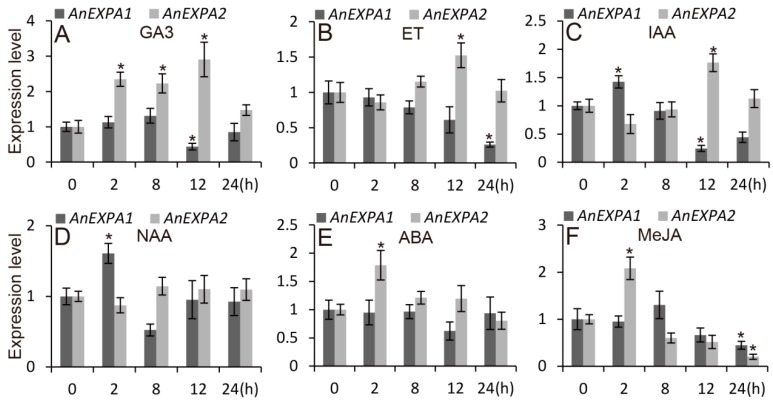
Expression levels of *AnEXPA1* and *AnEXPA2* during hormone treatments. (**A**–**F**) Relative expression of *AnEXPA1* and *AnEXPA2* in seedlings treated with 50 μM GA3, 50 μM ethephon (ET), 1 μM IAA, 2 μM NAA, 2 μM ABA, and 10 μM MeJA, respectively. Relative amounts were calculated and normalized with respect to actin RNA, and the experiment was repeated at least three times. Data correspond to the mean ± SE of three independent replicates. Symbols above the columns indicate significant differences (* *p* < 0.05).

**Figure 5 ijms-20-05255-f005:**
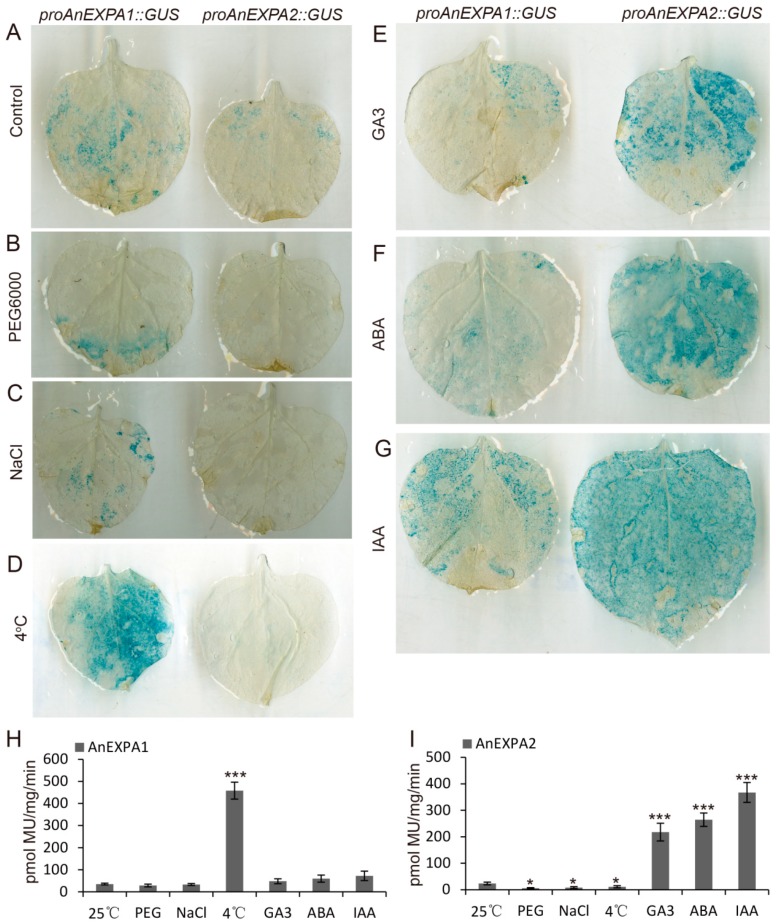
Histochemistry and β-glucuronidase (GUS) activity assays. (**A**–**G**) Under normal conditions, 40% PEG6000, 0.25 M NaCl, 50 μM GA3, 2 μM ABA, and 1 μM IAA treatments, the young tobacco leaves with proAnEXPA1::GUS and proAnEXPA2::GUS were dyed with X-Gluc solution. (**H**,**I**) After the above treatments, the GUS proteins of tobacco leaves with *proAnEXPA1::GUS* and *proAnEXPA2::GUS* were extracted and measured for GUS activity. Every treatment was repeated at least three times. Different symbols above the columns indicate significant differences (* *p* < 0.05, *** *p* < 0.001).

**Figure 6 ijms-20-05255-f006:**
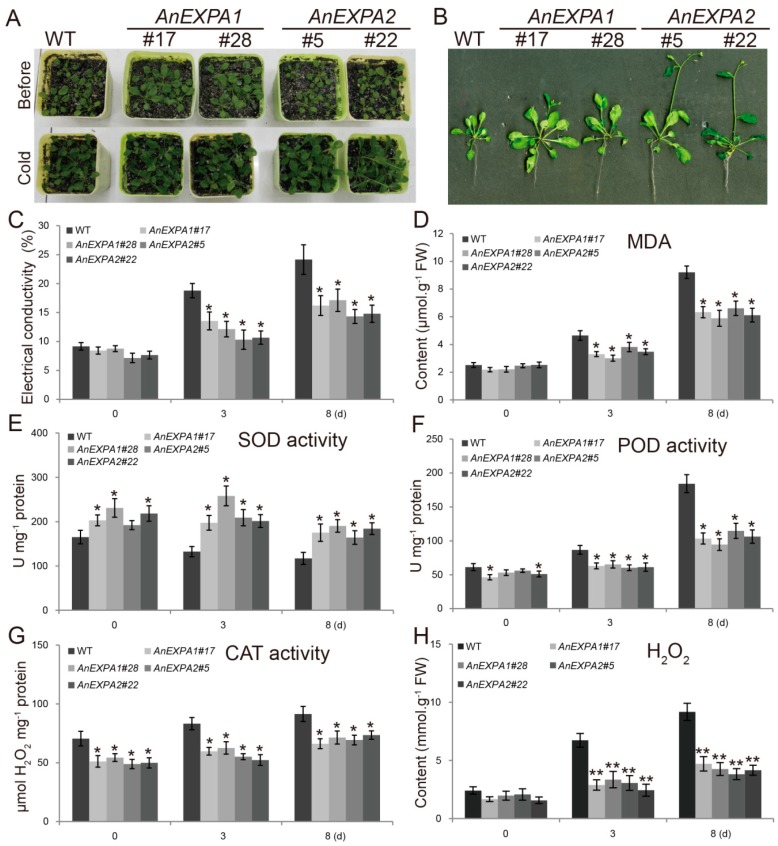
Analysis of cold tolerance in the *AnEXPA1* and *AnEXPA2* transgenic *Arabidopsis* plants. (**A**,**B**) Five-week-old transgenic seedlings were compared with non-transgenic plants before and after treatment at 6 °C for 8 d. (**C**–**H**) Comparison of electrical conductivity, malondialdehyde (MDA) content, activities of superoxide dismutase (SOD), peroxidase (POD), and catalase (CAT), and H_2_O_2_ content in the WT and transgenic plants at 6 °C for 8 days. Every transgenic line was measured at least three times. Different symbols above the columns indicate significant differences (* *p* < 0.05, ** *p* < 0.01). WT, non-transgenic *Arabidopsis* plants. #17 and #28, *AnEXPA1* transgenic lines. #5 and #22, *AnEXPA2* transgenic lines.

**Figure 7 ijms-20-05255-f007:**
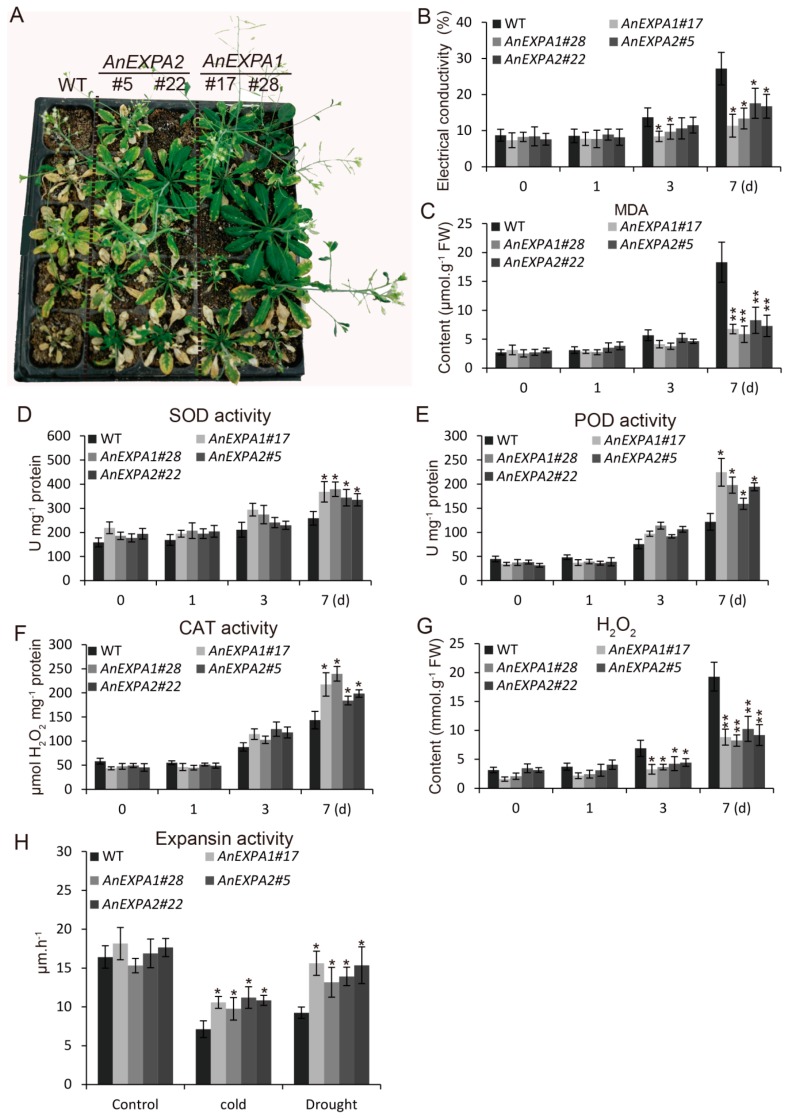
Analysis of drought tolerance in the *AnEXPA1* and *AnEXPA2* transgenic *Arabidopsis* plants. (**A**) Five-week-old transgenic plants under controlled water at normal temperature for 7 days. (**B**–**G**) Comparison of electrical conductivity, MDA content, the activities of SOD, POD, and CAT, and H_2_O_2_ content in the WT and transgenic *Arabidopsis* plants during controlled water treatment. (**H**) Expansin activity was analyzed by measuring the production of 4-methylumbelliferone (MU) under cold and drought stresses. Every transgenic line was measured at least three times. Different letters above the columns indicate significant differences (* *p* < 0.05, ** *p* < 0.01). WT, non-transgenic *Arabidopsis* plants. #17 and #28, *AnEXPA1* transgenic lines. #5 and #22, *AnEXPA2* transgenic lines.

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
