# Peer review of "Expression of Two α-Type Expansins from Ammopiptanthus nanus in Arabidopsis thaliana Enhance Tolerance to Cold and Drought Stresses"

_ijms, 2019, doi:10.3390/ijms20215255_

Round 1
Reviewer 1 Report
This new version of the manuscript has considerably improved, however english language and style must be improved throughout the manuscript before publication. Few more problems should still be addressed, detailed below as a reply to your answers to reviewers:
R: To better compare the differences of protein spatial structure, two templates Beta-expansin 1a (2hcz.1.A) and Pollen allergen (Phlp 1, 1n10.1.A) were used to build the 3D protein model. The native proteins and domain sequences of expansin and Pollen allergen are very similar, so the two templates were used to build 3D structure of AnEXPA1 and AnEXPA2 proteins.
the % aminoacid identity between the native proteins and domain sequences of expansin and Pollen allergen and AnEXPA1 and AnEXPA2 should be included in the text, as it will be useful for the readers to understand your criteria for selecting these two domains for templates.
R: Our two expansin genes were cloned from A. nanus, so some species from Leguminosae were chose. In addition, these study and classification about expansin gene family now were more comprehensive in Arabidopsis thaliana.Thus, expansins from Arabidopsis were also chose. Of course, expansins from other species similarity with our two expansin proteins (more than 60%) were also used in phylogenetic analysis.
this reply addresses only partially to my comment. the authors should clearly explain their selection criteria in the main text .
Figure 2 Numbers representing the level of clade support should be included in the phylogenetic tree
R: Here, our results suggest that the expression of two expansin genes showed tissue specificity and different abiotic stress responses.
This conclusion should be explicitly stated in the text.
line 205. this is not scientifically correct . analysing the expression of a promoter-GUS fusion will never tell you which element of the promoter is doing what. you will need to carry out a fine mutational analysis in order to find this out. this sentence must thus be changed.
R: Although the qRT-PCR analysis can show the changes in gene expression, do not directly show the basic expression level of genes. However, transient expression assay can directly express the effect of promoter on gene expression. Thus, GUS analysis is a complementary analysis of qRT-PCR and a test on basic expression level of genes.
it should be fully explained in the text why the authors chose to transform tobacco and assess the stress and hormone induction of the two genes in tobacco leaves, rather than on transgenic Arabidopsis plants. It should also be clearly stated if the experimental conditions used for this experiment were similar to those used for the qRT-PCR experiments.
line 242 change the sentence to "This could be due to constitutive high SOD protein levels, which may lead to low levels of ROS production.
Author Response
Reviewer 1
1. This new version of the manuscript has considerably improved, however english language and style must be improved throughout the manuscript before publication. Few more problems should still be addressed, detailed below as a reply to your answers to reviewers:
Response: We again requested LetPub for providing linguistic assistance, and amended some mistakes of English usage in our manuscript.
R: To better compare the differences of protein spatial structure, two templates Beta-expansin 1a (2hcz.1.A) and Pollen allergen (Phlp 1, 1n10.1.A) were used to build the 3D protein model. The native proteins and domain sequences of expansin and Pollen allergen are very similar, so the two templates were used to build 3D structure of AnEXPA1 and AnEXPA2 proteins.
The % aminoacid identity between the native proteins and domain sequences of expansin and Pollen allergen and AnEXPA1 and AnEXPA2 should be included in the text, as it will be useful for the readers to understand your criteria for selecting these two domains for templates.
Response: Thank you for your advice. We have added the sequence alignment of these two proteins with expansin and Pollen allergen in here (Figure A1).
2. R: Our two expansin genes were cloned from A. nanus, so some species from Leguminosae were chose. In addition, these study and classification about expansin gene family now were more comprehensive in Arabidopsis thaliana.Thus, expansins from Arabidopsis were also chose. Of course, expansins from other species similarity with our two expansin proteins (more than 60%) were also used in phylogenetic analysis.
This reply addresses only partially to my comment. the authors should clearly explain their selection criteria in the main text .
Response: Thank you for your advice. We have added our selection criteria in here.
3. Figure 2 Numbers representing the level of clade support should be included in the phylogenetic tree
Response: Thank you for your advice. We have added the numbers in the phylogenetic tree.
4. R: Here, our results suggest that the expression of two expansin genes showed tissue specificity and different abiotic stress responses.
This conclusion should be explicitly stated in the text.
Response: Thank you for your advice. We have explicitly stated in our manuscript.
5. Line 205. This is not scientifically correct. Analysing the expression of a promoter-GUS fusion will never tell you which element of the promoter is doing what. You will need to carry out a fine mutational analysis in order to find this out. This sentence must thus be changed.
Response: Thank you for your advice. We have amended this mistake in here.
6. R: Although the qRT-PCR analysis can show the changes in gene expression, do not directly show the basic expression level of genes. However, transient expression assay can directly express the effect of promoter on gene expression. Thus, GUS analysis is a complementary analysis of qRT-PCR and a test on basic expression level of genes.
It should be fully explained in the text why the authors chose to transform tobacco and assess the stress and hormone induction of the two genes in tobacco leaves, rather than on transgenic Arabidopsis plants. It should also be clearly stated if the experimental conditions used for this experiment were similar to those used for the qRT-PCR experiments.
Response: The tobacco leaves are usually used to study whether a gene promoter is response to different abiotic stress and plant hormones. Transgenic Arabidopsis plants are usually used to analyze the gene function. In other words, the gene CDS is transformed into Arabidopsis genome for obtaining the stable heredity transgenic plants, but the gene promoter is injected into tobacco leaves for studying the regulatory relationship of gene with abiotic stress and plant hormone. The two plants Thus, we selected tobacco leaves to study the regulatory relationship of gene with abiotic stress and plant hormone, and transgenic Arabidopsis plants to analyze the function of these two expansin genes in this study. In addition, the treatment conditions are the same and listed in the every picture description during treatment.
7. line 242 change the sentence to "This could be due to constitutive high SOD protein levels, which may lead to low levels of ROS production.
Response: Thank you for your advice. We have amended this sentence according to your advice.
Reviewer 2 Report
The figure legends still need to be carefully checked– The figure legends indicate that there are letters above columns to indicate significant differences, but there are only asterisks.
Author Response
Reviewer 2
The figure legends still need to be carefully checked– The figure legends indicate that there are letters above columns to indicate significant differences, but there are only asterisks.
Thank you for your suggestion. In many reports, the “*” above columns was used to replace letters during representation significant differences (*, P < 0.05; **, P < 0.01; ***, P < 0.001). Thus, we used “*” to replace letters. For example, in the following article:
Liu, YD., Zhang, L., Chen, LJ., Ma, H., Ruan, YY., Xu, T., Xu, CQ., He, Y., Qi, MF. Molecular cloning and expression of an encoding galactinol synthase gene (AnGolS1) in seedling of Ammopiptanthus nanus. Scientific Reports. 2016, 6:36113. Su, J., Hu, C., Yan, X., Jin, Y., Chen, Z., Guan, Q., et al. Expression of barley susiba2 transcription factor yields high-starch low-methane rice. Nature. 2015, 523(7562): 602-606.This manuscript is a resubmission of an earlier submission. The following is a list of the peer review reports and author responses from that submission.
Round 1
Reviewer 1 Report
This manuscript unfortunately has major problems in experimental design. In addition, not all experimental details are presented and the conclusions drawn by the authors are very often not supported by the data. I strongly recommend the authors to use an an example other papers already published on similar subjects and redesign their experiments accordingly. A detailed but not exhaustive list of comments follows.
104 it is not clear if the domains mentioned are only homology domains and /or they have a structural/functional significance. The authors should cite the references for both domains
Lines 97-114 this description is completely non accurate. Homology should be changed in as amino acid identity. The descritption of the aminoacids similarities/differences should be fully revised. I recommend the authors to read any good published paper containing a similar figure
115-116 this conclusion is not supported by the data. It should be omitted
Figure 1 legend domain 1 and 2 shooed be mentioned in the text and a reference should be provided
Lines 122-123 should be edited for clarity. Authors should mention what is the histidine, phenylalanine and aspartate residue (HFD) site and provide a reference
Lines 125-138. I don’t understand what is the need for thus manuscript to have a 3D model for the two expansions. In addition , why were 2hcz.1.A and Pollen allergen (Phlp 1) used as templates?
Lines 141-142. Which criteria were used to select the 59 expansins used in the tree?
Line 146. This conclusion is not supported by the data
Figure 2 . Numbers representing the level of clade support should be added to the phylogenetic tree. What software was used?
Lines what kind of software was used for predicting the promoter elements? Authors should provide a table with the sequences found, their positions and relative citations
Lines 179-181. This description lacks all necessary details.
Lines 182-185 these conclusions are not supportad by the data
Line 189 “cold “, “drought” and “salt” stress conditions should be defined in detail jn the main text
Lines 194-202 it will be important to describe the fact that not all stress induce the expression of the two genes with the same kinetics . Also the difference in the expression of the EXP genes in the three organs should be discussed. This applies also to the hormone treatments . It is puzzling/interesting that not all types of stress or hormones induce gene expression in the same way in different organs
Paragraph 2.6 it is not fully explain why the authors chose to transform tobacco and assess the stress and hormone induction of the two genes in tobacco leaves. In addition their conclusion are not fully supported the data. The GUS analysis should be compared with the results obtained in the qRT-PCR analysis. Were the experimental conditions used similar to those used in the qRT-PCR?
Paragraphs 2.7, 2.8 and figures 6 and 7 how many independent lines were generated ? The generation of transgenic plants is not described in detail. The experimental conditions are not explained in enough details. The different lines used seem to show different phenotypes but this is not correlated with different expression levels of the transgene(s). It is thus puzzling that different lines have different phentyoes. The data presented do not support the conclusions made by the authors .
Line 259 it is wild type, not wild. Many other errors can be found throughout the manuscript. The authors should use a standard Arabidopsis nomenclature.
Reviewer 2 Report
The manuscript is well designed and contains important findings. However, I have a couple of suggestions for improvementL
Please put statistical lettering on the bar (graphs)
Concise results.
Improve the discussion section. Use recent and relavent references.
Reviewer 3 Report
The information presented in the study is interesting and well presented. Because there is a lot of data presented in the manuscript be sure to carefully check that all references to tables and figures are correct and that all figure legends are correct.
Title
Looks fine
Abstract
Looks fine
Introduction
Looks fine
Material and Methods
Looks fine
Results and Discussion
Line 282 – make sure that discussion of data correctly refers to the figures that show that data (6A is cold not drought)
Figures and tables
Check figure legends – there are no letters above any columns just *.
References
Looks good